# Comparison of observational methods to identify and characterize post-COVID syndrome in the Netherlands using electronic health records and questionnaires

Isabelle Bos[1]*, Lisa Bosman[1], Rinske van den Hoek[1], Willemijn van Waarden[1], Matthijs S. Berends[2,3,4], Maarten S. Homburg[2], Tim Olde Hartman[5], Jean Muris[6], Lilian S. Peters[2,7], Bart Knottnerus[1], Karin S. Hek[1], Robert A. Verheij[1,8]

1 Nivel, Netherlands Institute for Health Services Research, Utrecht, The Netherlands, 2 Department of General Practice and Elderly Care Medicine, University Medical Center Groningen, Groningen, The Netherlands, 3 Certe Medical Diagnostics & Advice Foundation, Groningen, The Netherlands, 4 Department of Medical Microbiology and Infection Prevention, University Medical Center Groningen, Groningen, The Netherlands, 5 Department of Primary and Community Care, Radboud Institute of Health Sciences, Radboud University Medical Center, Nijmegen, The Netherlands, 6 Care and Public Health Research Institute (CAPHRI), Department of Family Medicine, Maastricht University, Maastricht, The Netherlands, 7 Amsterdam UMC, Department of Midwifery Science, Amsterdam Public Health Research Institute, Amsterdam, The Netherlands, 8 Tranzo, Tilburg School of Social and Behavioral Sciences, Tilburg University, Tilburg, The Netherlands

* i.bos@nivel.nl

**Data Availability Statement:** Data requests for usage of data from Nivel Primary Care Database or the Corona Survey Cohort may be submitted to the

## Abstract

### Background

Some of those infected with SARS-CoV-2 suffer from post-COVID syndrome (PCS). However, an uniform definition of PCS is lacking, causing uncertainty about the prevalence and nature of this syndrome. We aimed to improve understanding of PCS by operationalizing different classifications and to explore clinical subtypes.

### Methods

We used data from Nivel Primary Care database from 2019–2020 which consists of electronic health records (EHR) from general practices (GPs) combined with sociodemographic data for n = 10,313 individuals infected with the SARS-CoV-2. In addition, data from n = 276 individuals who had been infected with the SARS-CoV-2 in 2021, collected via a longitudinal survey, was used. In the GP-EHR data, we operationalized two classifications of PCS (based on symptoms and diagnosis recorded in GP-EHR data and healthcare utilization 3–12 months after acute infection) to calculate frequency and characteristics and compared this to the survey results. In a subgroup of the EHR data we conducted community detection analyses to explore clinical subtypes of PCS.

### Results

The frequency of PCS was 15% with on average 4.6 symptoms for which the GP was consulted using the narrow definition and 32% with on average 6.8 symptoms for the broad

applicable governance bodies via
gegevensaanvragen@nivel.nl. In this manuscript
data from Nivel Primary Care Database was linked
to data available at Statistics Netherlands and we
used the Microdata Platform of Statistics
Netherlands as a secure research environment. The
linked database is only available at the
Microdataplatform of Statistics Netherlands and
can be made accessible if certain conditions are
fulfilled.

**Funding:** This study was performed as part of the
Long COVID MM project which is financed by
ZonMw (https://www.zonmw.nl/en) under project
number 10430302110004. The following authors
received funding under this grant: IB, TOH, JM, LP
and KH. The funder did not play a role in the study
design, data collection, data analysis, decision to
publish or preparation of the manuscript.

**Competing interests:** The authors have declared
that no competing interests exist.

definition. Across all methods and classifications, the mean age of individuals with PCS was around 53 years and they were more often female. There were small sex differences in the type of symptoms and overall symptoms were persistent for 6 months. The community detection analysis revealed three possible clinical subtypes.

## Discussion

We showed that frequency rates of PCS differ between methods and data sources, but characteristics of the affected individuals are relatively stable. Overall, PCS is a heterogeneous syndrome affecting a substantial group of individuals who need adequate care. Future studies should focus on care trajectories and qualitative measures such as quality of life of individuals living with PCS.

## Introduction

Already in the first year of the global corona pandemic it became clear that a substantial part of the individuals infected with SARS-CoV-2 suffer from persistent symptoms, also called 'Post-COVID syndrome' (PCS) [1]. Although literature about PCS—or other related terminology like 'Long COVID' and 'Post-acute sequelae of COVID-19'- is now growing rapidly [2–6], there is still much debate about the prevalence and characteristics of people affected by PCS which is partly due to a lack of uniform definition [7]. The most commonly used definitions for PCS are those published between 2020 and 2021 by the World Health Organization (WHO) [8], the UK National Institute for Health and Care Excellence (NICE) [9] and the US Centers for Disease Control and Prevention (CDC) [10]. These three definitions already have discrepancies between them regarding the included symptoms and regarding the starting point of PCS: 4 weeks or 3 months after the infection. In addition, Chaichana and colleagues recently showed that of all the 295 studies conducted on PCS, more than 65% of the studies conducted used another definition than the three definitions listed above or no definition at all [7]. It is unclear what the direct impact of this heterogeneity in definitions among studies is on the outcomes like prevalence and characteristics of individuals with PCS.

Prevalence estimates reported so far on PCS vary widely between studies ranging from 13–80% [5, 6, 11, 12]. Besides the heterogeneity in used definitions the wide variety in reported prevalence estimates is also at least partly due to variations in the investigated population (e.g. hospitalized individuals due to the corona virus only vs. nationwide or vaccinated vs. non-vaccinated) [11, 13, 14]. Moreover, various studies are solely based on self-reported symptoms via questionnaires [6, 15, 16] and many also lack the ability to compare to control groups without persistent symptoms or without a SARS-COV-2 infection, leaving them unable to identify specific PCS symptoms and associations with for instance demographic factors [17, 18]. Altogether this makes it difficult to disentangle what PCS really is, what the impact of the syndrome is on societies and most importantly it hampers development and optimization of treatment strategies. In addition, besides clear diagnostic guidelines it would also aid to have more clarity on which symptoms often occur simultaneously in which type of individuals, also called clinical subtypes. Insights in clinical subtypes of PCS are useful in order to personalize clinical management strategies. There have been a few studies trying to identify clinical subtypes of PCS [19, 20] but these point in different directions so further research and validation is needed.

In this study, we therefore aimed to examine the direct impact of using different classifications and data sources on the estimated frequency of PCS and the characteristics of individuals suffering from it. By comparing patient reported outcomes measures (PROMs) to real world data from electronic health records (EHRs) we are able to demonstrate the differences in outcomes as every method and data source has its challenges, biases and advantages which should be taken into account [21–23]. With this study we aim to provide crucial novel insights into the differences which are needed for future research into PCS but also epidemiological research in general. In addition, operationalization of different classification is also useful for clinical practice as it could aid to further specify diagnostic guidelines. In addition, we aimed to explore whether we can identify clinical subtypes of PCS in routine healthcare data from electronic health records (EHR) using network analysis. To that extend we formulated the following research questions: 1) What is the impact of using different classifications and data sources on the frequency and characteristics of PCS?; and 2) Which clinical subtypes of PCS can be identified using routine healthcare data?

## Methods

The current study is part of the *Long COVID MM* (Long COVID Mixed Methods) project in which various methods are combined to provide insight into post-COVID syndrome. Metadata regarding this project can be found in the Health-RI COVID-19 data portal (https://covid19initiatives.health-ri.nl).

### Data sources

**GP-EHR database.** This database consists of electronic health records (EHR) from general practitioners (GP) and GP out-of-hours-services (OOH-services) combined on individual-level with demographic and socio-economic data. The EHR data from GPs and GP out-of-hours-services was obtained in November 2021 via Nivel Primary Care Data base (Nivel-PCD; approved under number NZR-00321.052) which uses an opt-out system permitted under the Dutch Medical Treatment Contracts Act (WGBO). Nivel-PCD covers about 10% of the Dutch population and the patients included form a representative sample of the population regarding age and sex [24]. General practices included in Nivel-PCD are located in all provinces of the Netherlands and the variation in urbanization level (scale 1–5) among practices is fairly similar as to all Dutch GP practices [24]. The GP-EHR data includes information on age, sex, prescriptions (coded via Anatomical Therapeutic Chemical classification, ATC), contacts, referrals, lab results and diagnosis or symptoms (International Classification of Primary Care-1, ICPC-1, coded). The OOH-services data include: contacts with diagnosis or symptoms (ICPC-1 coded), prescriptions (ATC-coded) and triage registration (ICPC-1 coded). All EHR data was pseudonymized and linked on individual-level by a trusted third party (ZorgTTP). The GP-EHR data was uploaded to the data platform of Statistics Netherlands and combined with demographic and socio-economic data collected at Statistic Netherlands including: age, sex, migration background, education level, household income and mortality data (date and cause of death). For the current study we used data from Nivel-PCD and Statistics Netherlands from 2019 and 2020 for n = 958,739 individuals in total. A flowchart of the included study population can be found in Supplementary Material (S1 Fig).

**Corona Survey Cohort.** As described elsewhere [25], this population-based cohort was initiated in May 2020 to study the long term effects of SARS-COV-2 infection and is based within Nivel-PCD. As part of the Long COVID MM study the initial cohort was extended with more participants and an additional follow-up survey. In short, n = 1851 individuals who had been flagged in their electronic patient file as having SARS-COV-2 infection by their GP

(ICPC code R83.03) were invited to participate between January and September 2021 in this study by their general practitioner (GP). Individuals were sent four surveys: direct after inclusion and after 3, 6 and 12 months. The surveys contained questions on symptoms, used health care and care experiences, quality of life, ability to work, vaccination status and selfcare. All participants signed informed consent forms allowing researchers to link the survey data to their GP EHR. This enables the unique opportunity to combine the survey data with EHR data on morbidities and prescriptions for this specific group. More details about the Corona Survey Cohort are described in a previous publication [25] and S1 Fig shows a flowchart of the included population.

**Ethical approval.** This project was conducted according to the Declaration of Helsinki and ethical approval was obtained from the medical ethics committee (METc) from the VU University Medical center Amsterdam for the longitudinal questionnaire component (METc protocol number 2020.0709) and from the METc of the University Medical Center Groningen for the electronic health records component (METc protocol number 2021/473). Conditions are fulfilled under which the use of electronic health records for research purposes in the Netherlands is allowed. Under these conditions, neither informed consent from study subjects nor approval by a medical ethics committee is obligatory for this type of observational studies, containing no directly identifiable data (art. 24 GDPR Implementation Act jo art. 9.2 sub j GDPR). All participants of the Corona Survey Cohort were adults (older than 18 years of age) and gave written informed consent before starting the survey and could additionally provide written informed consent for linkage of EHR data to questionnaire data.

## Classifications of post-COVID syndrome

**GP-EHR database.** In the GP-EHR database we conducted the following analyses to explore manners to operationalize different PCS classifications using routine healthcare data. First we selected individuals with SARS-CoV-2 infection (n = 10,313) based on the EHR data from Nivel-PCD (ICPC code R83.03) registered by their GPs directly, or who had been identified via a developed algorithm selecting patients based on symptoms and episode titles between April and June 2020 [26].

Each individual with SARS-CoV-2 infection was matched to four control individuals withoutSARS-CoV-2 infection. Matching to control patients was only used for the operationalization of the classifications. Controls were similar to individuals with SARS-CoV-2 infection in age and sex and were followed over the same period in the data as the individuals with SARS-CoV-2 infection to adjust for seasonal or circumstantial effects like lockdowns due to COVID-19 pandemic. Characteristics of the matched control group can be found in S1 Table. To validate the matching, we compared the matched control group to the total group with SARS-CoV-2 infection using t-tests for continuous outcome variables t-tests and Chi-square test for dichotomous variables (all p-values were > 0.05). To create a list of symptoms related to PCS, which is needed for the definition of PCS, we compared the ICPC codes recorded in the SARS-CoV-2 infection group, 3–12 months after infection, to the ICPC codes recorded in the same individuals a year before infection and to the ICPC codes recorded in the control group. Similar to the WHO definition we choose 3 months after SARS-CoV-2 infection as the cut-off point between acute SARS-CoV-2symptoms and PCS [8].

We created symptom lists following several steps. First, we ranked the ICPC codes by prevalence in the SARS-CoV-2 infection group nine months before and 3–12 months after infection. Thereafter, we calculated the difference in prevalence before and after infection and ranked these differences. We used the top 30 list of ICPC codes of which the prevalence was increased most after infection. We then excluded ICPC codes that had also increased in the

matched control group. This list was reviewed by four GP-researchers to exclude symptoms that were unlikely to be related to PCS. A total of n = 25 symptoms were included in the 'data-derived list'. In addition, we compiled a list of symptoms published by the WHO [11] and expanded this with symptoms reported by participants of the Corona Survey Cohort and a panel of 8 patients (age range: 32–75 years, 4 males and 4 females) who provide advice and feedback during the project. This 'patient reported list' included a total number of n = 37 symptoms. Furthermore, a GP (MH) and a medical microbiologist (MB) independently reviewed the entire list of ICPC codes for symptoms that could be related to acute SARS-CoV-2 infection and possibly also to PCS. This 'clinicians (acute) SARS-CoV-2 infection list' included n = 30 possible symptoms. We compared the symptoms on these three lists (i.e. 'data-derived list', 'patient reported list' and 'clinicians (acute) SARS-CoV-2 infection list') and symptoms that were included on at least two lists were considered 'core symptoms', while the remaining symptoms were considered 'additional symptoms' (S2 Table). We used the core and additional symptoms listed in S2 Table to classify the individuals with SARS-CoV-2 infection as having PCS using their EHR data according to a broad and narrow classification. According to the broad classification patients should have consulted the GP for at least one core symptom or at least two different additional symptoms, 3–12 months after SARS-CoV-2 infection. According to the narrow classification patients should have consulted the GP for at least two symptoms of which minimal one core symptom and at least two consultations for these symptoms at the GP. We created these two classifications with current literature and definitions of PCS in mind (e.g. but not limited to [8, 14, 17]), input from GPs (JM, TOH, BK) and the involved researchers. We used these two classifications because there is currently no uniform definition and we aimed to investigate the influence of using different classifications. The broad classification was created to be inclusive to possible heterogeneity of the syndrome and the narrow classification to depend more on care usage and the core symptoms.

**Corona Survey Cohort.** Of the total number of participants in the Corona Survey Cohort (n = 442), n = 276 (62%) participants were selected for the current analysis as they completed the first questionnaire within 3 months after SARS-CoV-2 infection (between January–September 2021) and could answer the questions regarding acute symptoms more accurately. Individuals were classified as having PCS when they reported at least one symptom, from a selected list of symptoms, three months after the SARS-CoV-2 infection and experienced discomfort in their daily living (first survey) or reported not be recovered after the initial SARS-CoV-2 infection (second survey). Individuals were classified as non-PCS (n = 93) when they reported not to experience discomfort or reported to be recovered. Individuals were classified as 'unknown' (n = 92) when relevant data to classify individuals as PCS or non-PCS was missing or when there was a discrepancy in the answers to the questionnaire (i.e. report no symptoms, but also report not to be recovered).

**Outcomes and covariates.** Age categories were divided into: children and adolescents (age 0–23 years of age), adults (24–70 years of age), elderly ($\geq$ 70 years of age). Migration background was dichotomized as: both parents were born in the Netherlands (0) and at least one parent is not born in the Netherlands (1). Education level was divided into low (primary school of pre-vocational education), medium (secondary or vocational education) and high (professional higher education or university) education level. Income level was only available in the GP-EHR database and was divided according to standardized household income in the Netherlands into low (0–40 percentile), medium (40–80 percentile) and high (>80 percentile). GP consultations were defined as long, medium and short consultations including consultations by phone and email and long and short visitations. Long and short consultations with the nurse practitioner were also included.

## Statistical analysis

We used descriptive statistics to describe the sample characteristics of the PCS patients in the combined EHR database and the Corona Survey Cohort. For continuous outcome variables t-tests were used to compare groups and for dichotomous outcome variables Chi-square test were used. In a subgroup of the EHR database we performed a network analysis, Louvain Community Detection [27], to identify symptoms that often co-occur in individuals with PCS. For these analyses we only included individuals who consulted the GP for two different symptoms (n = 1,503) and we used the R-package 'igraph' for visualization. A community was included in the network when at least 1% of individuals have this combination of symptoms and we used a cut-off of >0.3 on the modularity score to ensure the quality of the communities and network [28]. We then classified the 1503 individuals with at least two symptoms in one community and described the demographic characteristics of these individuals. For all analyses, p-values below 0.001 were deemed statistically significant. Statistical analyses were performed in STATA (version 16.1) and R (version 4.1.3).

## Results

In the GP-EHR database we selected n = 10,313 individuals who were all infected with the SARS-CoV-2 virus in 2020. Of these individuals, n = 452 (4.3%) were hospitalized due to SARS-CoV-2 infection during the acute phase (0–3 months after infection). Table 1 describes the characteristics of these individuals, classified according to the broad and narrow classifications of PCS. The selection of the Corona Survey Cohort we used for the current analysis included n = 276 individuals who had been infected by SARS-CoV-2 virus. Of these individuals n = 18 (6.6%) had been hospitalized during the acute phase (0–3 months after infection). Table 2 describes the characteristics of the individuals from the Corona Survey Cohort. The percentages of individuals classified as having PCS ranged from 15–33% depending on the classification and data source (Tables 1 and 2).

### GP-EHR data: Demographics and other characteristics using broad and narrow PCS classifications

In the GP-EHR data comparisons were made between individuals in the PCS group and the non-PCS group according to the broad and narrow classifications (Table 1). Results of the comparisons were similar for both classifications and therefore we only mention the results using the narrow definition in the text. Individuals with PCS were more often female (69% vs. 57%, p≤0.001) and were older (53.4 vs. 51.1 years, p≤0.001) compared to the non-PCS group. There were significantly fewer children in the PCS group compared to the non-PCS group and more adults and elderly in the PCS groups (Table 1). There was no difference between the PCS group and the non-PCS group in education level, household income or migration background. The average number of symptoms for which the GP was consulted by individuals with PCS was 6.8 (SD 5.4) symptoms per patient versus 0.9 (SD 1.8) in the non-PCS group (p≤0.001). The average number of GP consultations was, by definition, higher in the PCS group compared to the non-PCS group (5.5 vs. 0.8 consultations, p≤0.001).

### Corona Survey Cohort: Demographics and other characteristics in PCS and non-PCS group

In the Corona Survey Cohort (Table 2) we compared the PCS group to the non-PCS group. The characteristics of the unknown group (n = 92) can be found in S3 Table. Unlike in the GP-EHR data, there was no significant difference in age and sex between the PCS group and

**Table 1. Characteristics of GP-EHR data.**

| | PCS group–broad definition | Non PCS group–broad definition | PCS group—narrow definition | Non PCS group–narrow definition | Total group individuals with SARS-CoV-2 infection |
|---|---|---|---|---|---|
| n (% of total SARS-CoV-2 infected group) | 3,333 (32.3) | 6,980 (67.7) | 1,533 (14.9) | 8,780 (85.1) | 10,313 (100) |
| Age (mean, SD) | 53.0 (18.3)** | 50.8 (20.1) | 53.4 (18.3)** | 51.2 (19.8) | 51.5 (19.6) |
| Children and adolescents, n (%) | 380 (5.4) | 380 (5.4) | 22 (1.4)** | 429 (4.9) | 451 (4.4) |
| Adults, n (%) | 5,326 (76.3) | 5,326 (76.3) | 1,206 (78.7)** | 6,758 (77.0) | 7,964 (77.2) |
| Elderly, n (%) | 1,274 (18.3) | 1,274 (18.3) | 305 (19.9)** | 1,593 (18.1) | 1,898 (18.4) |
| Male, n (%) | 1,127 (33.8%)** | 3,111 (44.6) | 472 (30.8)** | 3,766 (42.9) | 4,238 (41.1) |
| Low education level categories | | | | | |
| Low, n (%) | 1,854 (26.6) | 1,854 (26.6) | 412 (26.9) | 2,318 (26.4) | 2,730 (26.5) |
| Medium, n (%) | 1,179 (16.9) | 1,179 (16.9) | 288 (18.8) | 1,470 (16.7) | 1,758 (17.0) |
| High, n (%) | 1,302 (18.7) | 1,302 (18.7) | 255 (16.6) | 1,661 (18.9) | 1,916 (18.6) |
| Unknown, n (%) | 2,645 (37.9) | 2,645 (37.9) | 578 (37.7) | 3,331 (37.9) | 3,909 (37.9) |
| Household income categories | | | | | |
| Low, n (%) | 2,687 (38.5) | 2,687 (38.5) | 628 (41.0) | 3,357 (38.2) | 3,985 (38.6) |
| Medium, n (%) | 2,688 (38.5) | 2,688 (38.5) | 567 (37.0) | 3,382 (38.5) | 3,949 (38.3) |
| High, n (%) | 1,305 (18.7) | 1,305 (18.7) | 273 (17.8) | 1,662 (18.9) | 1,935 (18.8) |
| Unknown, n (%) | 300 (4.3) | 300 (4.3) | 65 (4.2) | 379 (4.3) | 444 (4.3) |
| Migration background, n (%) | 716 (21.5) | 1377 (19.7) | 353 (23.0) | 1740 (19.8) | 2.093 (20.3) |
| No. of GP consultations 3–12 months after infection (mean, SD) | 3.8 (3.5)** | 0.4 (1.1) | 5.5 (4.0)** | 0.8 (1.6) | 1.5 (2.7) |
| No. of different symptoms 3–12 months after infection (mean, SD) | 4.6 (4.6)** | 0.4 (1.2) | 6.8 (5.4)** | 0.9 (1.8) | 1.8 (3.4) |

*p≤0.005 compared to the non-PCS group

**p≤0.001 compared to the non-PCS group; For continuous variables t-tests were used and for dichotomous variables Chi-square analysis were conducted;

GP = general practitioner. SD = standard deviation

the non-PCS group. Individuals in the PCS group more often had a lower education level (p≤0.001) compared to the non-PCS group. The average number of self-reported symptoms in the PCS group was significantly higher compared to the non-PCS group at 3 months after infection (9.2 vs. 2.8; p≤0.001) and also 6 months after infection (7.2 vs. 2.1; p≤0.001). Twenty-one (23%) individuals with PCS reported that they are working less or stopped working due to PCS symptoms after 3 months and 11 (16%) after 6 months (Table 2).

## GP-EHR data: Frequency of symptoms stratified by sex

Fig 1 shows the frequencies of patients that visit the GP for a particular symptom 3–12 months after the SARS-CoV-2 infection stratified by PCS classification and sex for the top 10 most prevalent core symptoms based on the narrow definition. For all symptoms we found that males consulted their GP less often for these symptoms compared to females. The most prevalent symptoms in females were psychological symptoms (22–25%) including anxiety and depression, while respiratory symptoms (15–19%) like coughing or dyspnea were most prevalent in males.

**Table 2. Characteristics of the different groups within the Corona Survey Cohort.**

| | PCS | Non–PCS | p-value |
|---|---|---|---|
| n (% of total Corona Survey Cohort included population) | 91 (33) | 93 (34) | |
| Age (mean, SD) | 53.6 (13.6) | 52.3 (12.1) | 0.479 |
| Male, n (%) | 36 (39.6) | 31 (33.3) | 0.380 |
| Level of education | | | ≤0.001 |
|  Low, n (%) | 31 (34.1) | 10 (10.8) | |
|  Medium, n (%) | 27 (29.7) | 33 (35.5) | |
|  High, n (%) | 24 (26.4) | 48 (51.6) | |
|  Unknown, n (%) | 9 (9.9) | 2 (2.2) | |
| Migration background. n (%) | 8 (8.8) | 4 (4.3) | 0.422 |
| No. of (self-reported) symptoms | | | |
|  After 3 months | 9.2 (4.8) | 2.8 (3.4) | ≤0.001 |
|  After 6 months | 8.6 (5.4) | 2.8 (3.7) | ≤0.001 |
| Individuals who are working less or stopped working | | | |
|  After 3 months (n, % of total n = 184) | 21 (23.1) | 0 (0.0) | ≤0.001 |
|  After 6 months (n, % of total n = 147) | 11 (15.5) | 1 (1.6) | 0.005 |

For continuous variables t-tests were used and for dichotomous variables Chi-square analysis were conducted; PCS = post COVID syndrome, SD = standard deviation.

## Corona Survey Cohort: Frequency of symptoms over time in males and females

Fig 2 shows the frequency of symptoms for the PCS patients in the Corona Survey Cohort at 3 and 6 months, stratified by gender. The top 10 most prevalent symptoms at 3 months are shown. Overall, symptom frequencies are considerably higher in the Corona Survey Cohort compared to the GP-EHR data and different symptoms are reported (Fig 1). In the Corona Survey Cohort, the most prevalent and persistent symptom was fatigue in both males (3 months: 89%, 6 months 78%) and females (3 months: 89%, 6 months: 86%). Similar to the

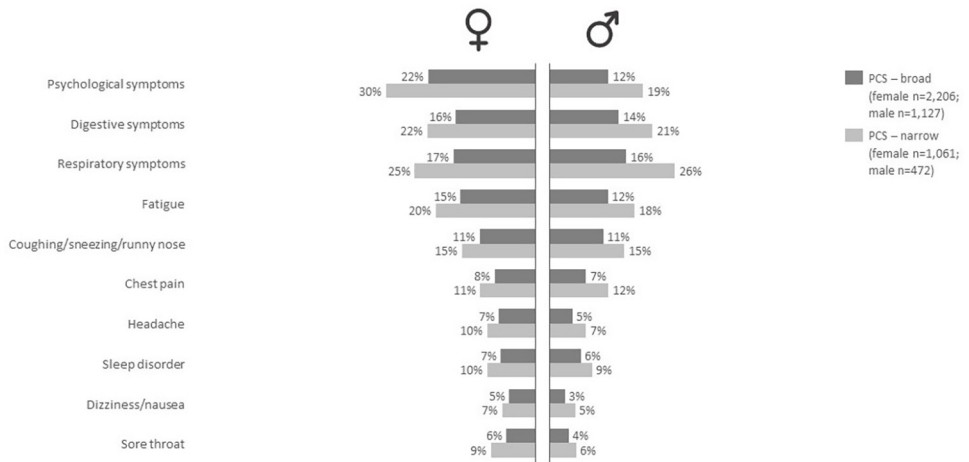

**Fig 1. Frequency of symptoms in the GP-EHR data stratified by sex.** Barplot showing the frequency of occurrence of category of symptoms in the GP-EHR data for the broad (dark grey) and narrow classifications (light grey) stratified for females (left) and males (right). Top 10 symptoms was based on the narrow definition.

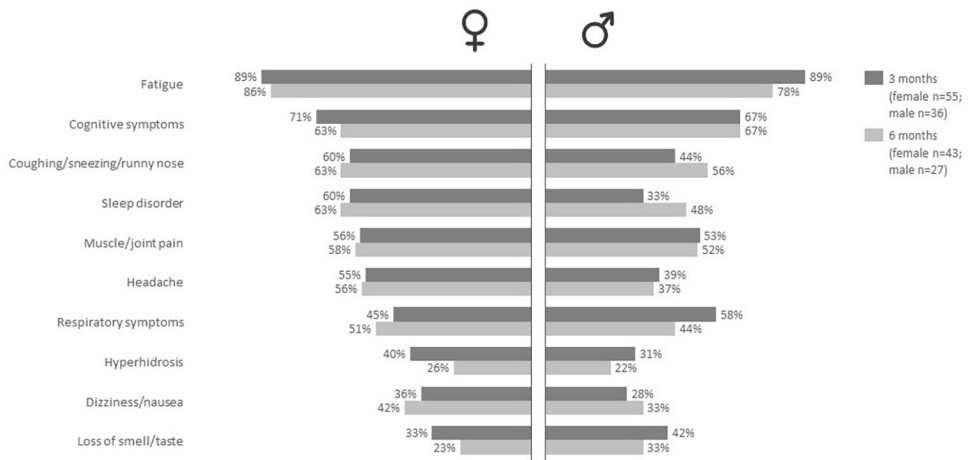

**Fig 2. Frequency of symptoms in the Corona Survey Cohort and 3 and 6 months after infections, stratified by sex.**
Barplot showing the frequency of self-report symptoms in the Corona Survey Cohort at 3 months (dark grey) and 6 months (light grey) after infection stratified for females (left) and males (right). The top 10 most prevalent symptoms at 3 months are shown in the graph.

GP-EHR data, males reported less symptoms than females and males more often reported respiratory symptoms compared to females at 3 months (58% vs. 45%), while this is opposite at 6 months (44% vs. 51%). Overall, in females the reported frequencies decreased for four symptoms while it increased for six symptoms. In males we found that the frequency decreased or stayed the same for seven symptoms while it increased for three symptoms (Fig 2).

## GP-EHR data: Community detection analyses to explore clinical subtypes

To explore possible clinical subtypes of PCS community detection analyses were performed in a subgroup of individuals in the GP-EHR data (n = 1,503) who visited the GP for at least two different symptoms. Logically, this is roughly the same group of individuals as the PCS group defined by the narrow classification (n = 1,533) except that we did not include data from individuals who visited the GP OOH-services (n = 30) for these analysis. Fig 3 shows the results of the community detection of the combination of symptoms that often occur together. We identified a network with a modularity score (possible range: -0.5 to 1.0) of 0.302 indicating a network with average strength in which three communities with symptoms were identified (Fig 3). Community A includes psychological and generalized symptoms and was statistically significant (p = 0.045), Community B includes cardiorespiratory symptoms (p = 0.494) and Community C includes gastrointestinal symptoms (p = 0.617). The communities are solely based on symptoms that co-occur and not on how many individuals have only these combinations of symptoms. Therefore, we subsequently analyzed how many individuals of this subgroup (n = 1,503) could be classified as experiencing this combination of symptoms (i.e. having at least two symptoms within one community). When classifying the group into individuals with at least two 'community symptoms' we found that n = 248 (17%) had symptoms across communities and n = 458 (30%) had symptoms that were not included in the network. In addition, there were individuals with only a single 'community symptom', n = 360 (24%) in community A, n = 126 (8%) in community B and n = 150 (10%) in community C. Table 3 shows the characteristics of the individuals who could be classified as experiencing distinct community symptoms. The group with neuro-respiratory symptoms (Community A) was the largest group (n = 109, 7%) and often females with an average age of 54.2. Thirty-two (2%) individuals

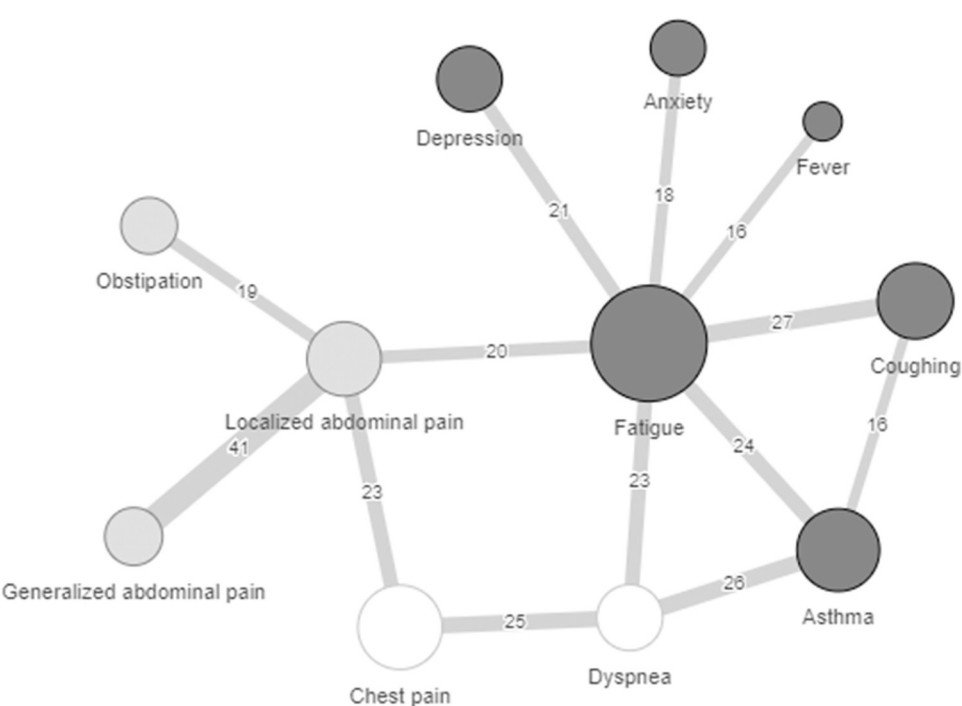

**Fig 3. Community detection analyses of symptoms in subgroup of individuals with PCS.** Communities of symptoms that co-occur that were detected in a subgroup (n = 1503) individuals of the GP-EHR database. Three communities were detected which are displayed with different colors: Community A including psychological-generalized symptoms (dark grey), Community B including gastrointestinal symptoms (light grey) and Community C including cardiorespiratory symptoms (white). The size of the circles shows how often symptoms occur in the data with bigger circles occurring more frequent. The numbers indicate how often symptoms co-occur.

experienced only symptoms from community B which are gastrointestinal symptoms. These individuals were younger (mean age 51.2) and included the lowest percentages of females (68%). The group with cardiopulmonary symptoms was the smallest including n = 18 (1%) who had an average age of 55.3 years. The percentages of individuals with a migration background was similar across the community groups (22–24%; Table 3).

## Discussion

This study describes the extent to which classifications and data collection methods are associated with the frequency of post-COVID syndrome as well as its constituting symptoms. By combining the results of the analyses of GP-EHR data and longitudinal questionnaire the main findings are: 1) the frequency of PCS among individuals infected by the SARS-CoV-2 between April–July 2020 in the Netherlands, ranged from 15–33% depending on the classification and data source used; 2) individuals with PCS were on average 53 years old and more often female; 3) individuals with PCS consulted the GP most often with psychological problems, while fatigue was the most often self-reported symptom; 4) three communities of possible related PCS symptoms were identified but require further examination and validation to define clinical subtypes of PCS.

Thus far worldwide prevalence rates of PCS vary widely between studies depending on populations, methods and definitions. Two recent meta-analysis regarding the prevalence of PCS for instance showed that prevalence rates were higher among individuals who were hospitalized during the acute phase compared nonhospitalized populations [14, 17]. Our study

**Table 3. Characteristics of patients in different symptom clusters.**

| | Group with Community A–psychological-generalized symptoms | Group with Community B–gastrointestinal symptoms | Group with Community C–cardiorespiratory symptoms |
|---|---|---|---|
| n (% of total group included in network analysis. n = 1503) with at least two community symptoms | 109 (7%) | 34 (2%) | 18 (1%) |
| Age (SD) | 54.2 (20.0) | 51.2 (23.2) | 55.3 (21.1) |
| Female, n (%) | 83 (76%) | 23 (68%) | 13 (72%) |
| Low education level. n (%) | | | |
| High | 26 (24%) | n≤10 | n≤10 |
| Medium | 26 (24%) | 14 (41%) | n≤10 |
| Low | 17 (16%) | 11 (32%) | n≤10 |
| Unknown | 40 (36%) | n≤10 | n≤10 |
| Migration background. n (%) | 26 (24%) | n≤10 | n≤10 |

includes both nonhospitalized and hospitalized individuals in the GP-EHR data (4% hospitalized) as well as in the Corona Survey cohort (7% hospitalized), although our group of hospitalized individuals are small in comparison to hospitalization rates due to acute SARS-CoV-2 infection in the Netherlands at that time [29, 30]. This might cause for a slight underestimation of the PCS frequency in our study. Another important factor influencing the variety among prevalence rates reported is whether a control group and comparisons on individual level with pre-COVID situation regarding symptoms and comorbidities has been included [14]. The few studies that have also included this crucial correction for a control group, like our study, generally report lower prevalence rates, similar to our results [6, 31]. Other obvious but central differences between studies on prevalence rates are whether PCS is defined based on self-report and whether patients are included based on only one symptom or on multiple symptoms. Our results underline and clarify the influence of these factors and the impact it has on the characterization of the group individuals suffering from PCS as we compared a broad (minimal 1 symptom) and narrow (minimal 2 symptoms and multiple consultations) classifications in the EHR data and the self-report data from the Corona Survey Cohort. Besides the obvious influence the narrow and broad definition have on the size of the PCS group it did not influence the characteristics (i.e. age, gender, migration background) of the individuals included. On the other hand, when comparing the PCS group in self-report survey data (Corona Survey Cohort) we did find a noteworthy difference in the level of education between the PCS and the non-PCS group which we did not find in the EHR data. In the survey data we found a higher percentage of individuals with a low education in the PCS group compared to the non-PCS group and the total group. This finding is in line with a German study which also showed that higher level of education was associated with a lower risk of PCS [13]. The lack of association in the GP-EHR data could be due to the large number of individuals for whom the level of education was unknown (38%), although the distribution among the education categories in the group of

individuals in the PCS group for whom this is known (62%) is similar to the distribution in the total COVID group. Future studies should further investigate the association between PCS and education level to validate our findings.

In general, findings thus far published regarding sex and PCS are quite consistent and also in line with our results as most studies report a higher occurrence of PCS in females compared to males [3, 17, 32, 33]. In addition, we also found that females report or seek help for different PCS symptoms than males. Females with PCS more often consult the GP for mental health symptoms, while males consult the GP most often for respiratory symptoms. A previous study also reported sex differences in relation to PCS symptoms but only included somatic symptoms and no psychological or mental symptoms [6]. Nevertheless, our results are in line with a large body of literature showing that males are less likely to seek medical help, in particular for mental health problems [34]. In general the most prevalent PCS symptoms for which the GP is consulted are psychological symptoms including anxiety and depressed mood, digestive symptoms including diarrhea and obstipation and respiratory symptoms including dyspnea and trouble breathing. Surprisingly, fatigue is not the most often reported symptom at the GP while a meta-analysis reported that this is the most common symptom of PCS [35]. Yet when focusing on self-reported symptoms in our survey data, fatigue is found to be the most common symptom. This emphasizes the differences between using routine healthcare registry data and self-report data which has been reported before [22, 36, 37], but requires further examination in relation to PCS.

We found that in the Corona Survey Cohort 23% of the individuals with PCS stopped working or worked less after three months compared to 0% in the non-PCS group. After six months 11% of individuals with PCS stopped working or worked less. It has been reported in other studies as well that work ability can be severely affected by PCS which may have large consequences at individual but also on societal level [38–40]. Tailored interventions for PCS in relation to work focusing on management of symptoms, impact on work ability, possible workplace adjustments and job modifications should be considered. On the other hand we found that there is also a substantial group of individuals with PCS (77%) who were able to maintain work ability which may be related to their type of work or to the type of healthcare provided to cope or recover from PCS. Large scaled analysis focusing on healthcare utilization patterns in PCS combined with outcome measures such as work (dis)ability are needed to further assess the relationship between PCS and work. In addition, further research which follows a patient group as well as a reference group, in this case the non-PCS group, over a longer period of time and also after the COVID-19 pandemic is important as well. In this current study we only focused on a relatively short period and found very little changes regarding work in the non-PCS group (0% at 3 months and 1% at 6 months) which could be the result of an inclusion bias or related to the fact that 'experiencing discomfort in daily living' was part of the definition of PCS.

To examine whether we could use machine learning to identify specific clusters of symptoms that often co-occur we performed a community detection analysis. We identified three communities which only partly overlap with clusters identified in a previous study which identified clusters across different SARS-CoV-2 variants [20]. Similar to our findings, Canas and colleagues (2022) also identified a cluster with mainly cardiorespiratory symptoms which was associated with the wild-type variant of the virus (i.e. first stages of the pandemic). The other clusters and communities associated with the early variant of the virus were however different from our findings [20]. Another study using electronic health records and a data-driven approach to identify symptom patterns in PCS identified two groups based on the prevalence of symptoms rather than the combination of co-occurring symptoms [41]. In addition literature studies on symptoms that often co-occur do however also often show a cardiorespiratory

cluster [42–44], a generalized-mood or neuropsychiatric cluster [45, 46], and a gastrointestinal cluster [18, 47]. Overall it has become clear that not all literature points in the same direction as also other clusters have been mentioned depending on population, virus variant and clustering method [48]. Our results are not conclusive on the clinical subtypes and should be interpreted with caution as groups were small and there were many individuals with symptoms in multiple clusters or other combinations of symptoms not identified by the analyses. It is also important to mention that the community detection analyses were conducted in a subgroup of individuals whom consulted the GP for various symptoms, which could indicate a more severe phenotype of PCS and might not occur in all individuals with PCS. Future studies, perhaps using data with biological and continuous parameters, should validate and further examine possible phenotypes of PCS as EHR might be not be well suited for these types of analysis due to the categorical coding and registration limitations [21].

The major strengths of this study lie in generalizability of the findings and the combination of methods which serve as an internal validation of the findings. Our findings from the GP-EHR data are generalizable to the Dutch population as this database includes a representative sample of the Dutch population. In addition, data from before the COVID-19 pandemic regarding healthcare usage and matched controls to define our PCS groups and compared to two reference groups (non-PCS and non-COVID). In addition, the combination of methods (using EHR data and surveys) allows for internal validation and interpretation of the results and provides a unique opportunity to compare frequency rates and symptoms reported in self-report data and routine healthcare data. Also, by using different sources of input (data-driven, list of WHO and experts) to create a list with core and additional symptoms added rigor to our study. However, also some limitations of this study should also be acknowledged. First, we identified patients as having had a SARS-CoV-2 infection when they visited their GP with COVID related symptoms and not by including all patients who were tested positive for the virus by the national testing authorities as public testing was not yet available during this time period. This limitation also applies to the control cases of whom we assumed that they did were infected with SARS-CoV-2 at the time of the study. Second, in the survey data (Corona Survey Cohort) there may be a selection bias as individuals who are experiencing persistent symptoms may be more likely to complete questionnaires compared to individuals not experiencing symptoms. In addition, there was a subgroup (the unknown group, 33%) of the Corona Survey Cohort which were not able to classify as having PCS or not due to missing data or discrepancies in the data and was therefore excluded from the analysis. These are insuperable biases when using survey data which should be taken into account when interpreting the results. Lastly, in this paper we only focused on individuals who were infected with SARS-CoV-2 in the first period of the COVID-19 pandemic and therefore not all variants of the virus are included. In future studies it would be possible to examine the relationship between virus variants and PCS.

## Conclusions

In conclusion, our results indicate how classifications and the choice of data sources may affect the frequency of PCS and the characteristics of the individuals affected by it as well as symptoms that are regarded as part of it. Frequency rates differ between methods and data sources (15–33%). Using the EHR cohort characteristics of the PCS population were stable across methods as we found that is mostly affects middle-aged females. In the survey cohort however, the PCS group did not differ from the non-PCS group regarding age and sex. The insights from this study form a solid basis for subsequent analyses on quality of life, care trajectories and risk factors for developing PCS. These analyses have been conducted in parallel studies to improve understanding and care for individuals with PCS, which is desperately needed.

## Supporting information

**S1 Fig. Flowchart of study populations.** Schematic overview of the included study populations of the GP-EHR cohort and the Corona Survey Cohort.
(PNG)

**S1 Table. Characteristics of matched control group.** Demographic and socioeconomic characteristics of the matched reference group that was used for classification.
(DOCX)

**S2 Table. List of core and additional symptoms and diagnosis used in GP-EHR cohort to classify PCS.** Symptoms and diagnosis are based on ICPC coding and are classified as core or additional symptom. Symptoms and diagnosis are also classified in categories which are used in Fig 1.
(DOCX)

**S3 Table. Characteristics of the unknown group in Corona Survey Cohort.** Demographic and socioeconomic characteristics of individuals in the Corona Survey Cohort who could not be classified and were therefore excluded from the analyses.
(DOCX)

## Author Contributions

**Conceptualization:** Isabelle Bos, Lisa Bosman, Rinske van den Hoek, Maarten S. Homburg, Tim Olde Hartman, Jean Muris, Lilian S. Peters, Bart Knottnerus, Karin S. Hek, Robert A. Verheij.

**Data curation:** Lisa Bosman, Rinske van den Hoek, Willemijn van Waarden, Karin S. Hek.

**Formal analysis:** Rinske van den Hoek, Willemijn van Waarden.

**Funding acquisition:** Isabelle Bos, Karin S. Hek, Robert A. Verheij.

**Investigation:** Isabelle Bos, Lisa Bosman.

**Methodology:** Isabelle Bos, Lisa Bosman, Matthijs S. Berends, Karin S. Hek.

**Project administration:** Isabelle Bos, Karin S. Hek.

**Resources:** Tim Olde Hartman.

**Supervision:** Lilian S. Peters, Robert A. Verheij.

**Validation:** Matthijs S. Berends, Maarten S. Homburg.

**Visualization:** Willemijn van Waarden.

**Writing – original draft:** Isabelle Bos.

**Writing – review & editing:** Lisa Bosman, Rinske van den Hoek, Willemijn van Waarden, Matthijs S. Berends, Maarten S. Homburg, Tim Olde Hartman, Jean Muris, Lilian S. Peters, Bart Knottnerus, Karin S. Hek, Robert A. Verheij.

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
