## [Decision Letter · Decision Letter 0]

30 Aug 2024

PONE-D-24-29474Comparison of observational methods to indentify and characterize Post-COVID syndrome in the Netherlands using electronic health records and questionnairesPLOS ONE

Dear Dr. Bos,

Thank you for submitting your manuscript to PLOS ONE. After careful consideration, we feel that it has merit but does not fully meet PLOS ONE’s publication criteria as it currently stands. Therefore, we invite you to submit a revised version of the manuscript that addresses the points raised during the review process.

We look forward to receiving your revised manuscript.

Kind regards,

Dong Keon Yon, MD, FACAAI, FAAAAI

Academic Editor

PLOS ONE

“This study was performed as part of the Long COVID MM project which is financed by ZonMw under project number 10430302110004.”

4. In this instance it seems there may be acceptable restrictions in place that prevent the public sharing of your minimal data. However, in line with our goal of ensuring long-term data availability to all interested researchers, PLOS’ Data Policy states that authors cannot be the sole named individuals responsible for ensuring data access (http://journals.plos.org/plosone/s/data-availability#loc-acceptable-data-sharing-methods).

5. We note that you have indicated that there are restrictions to data sharing for this study. PLOS only allows data to be available upon request if there are legal or ethical restrictions on sharing data publicly. For more information on unacceptable data access restrictions, please see http://journals.plos.org/plosone/s/data-availability#loc-unacceptable-data-access-restrictions.

7. Please amend either the title on the online submission form (via Edit Submission) or the title in the manuscript so that they are identical.

Additional Editor Comments:

Please address the excellent comments from the reviewers.

Reviewers' comments:

Reviewer's Responses to Questions

**Comments to the Author**

1. Is the manuscript technically sound, and do the data support the conclusions?

Reviewer #1: Partly

Reviewer #2: Yes

2. Has the statistical analysis been performed appropriately and rigorously? 

Reviewer #1: Yes

Reviewer #2: Yes

3. Have the authors made all data underlying the findings in their manuscript fully available?

Reviewer #1: No

Reviewer #2: Yes

4. Is the manuscript presented in an intelligible fashion and written in standard English?

Reviewer #1: Yes

Reviewer #2: Yes

5. Review Comments to the Author

Reviewer #1: This is an exploratory and descriptive paper investigating some different ways of classifying post-COVID syndrome (PCS) using a dataset derived from Electronic Health Records (EHRs) and a survey of a small subset of individuals selected from those electronic health records. Overall, the standard of English is OK and the methodology, given my somewhat limited knowledge about the technique used to identify symptom clusters, is mostly fine. However, I have one major concern about the Corona virus survey, and the paper could be improved in terms of clarity of reporting, and the theoretical basis behind the paper.

The major issue with the Corona Virus survey is that it was not possible to determine for around a third of the sample whether or not they had PCS. Given this, and what is probably a low response rate I am not convinced that estimates of frequency from this sample are particularly meaningful.

There is also lack of clarity and transparency around the sample size used in analyses. As far as I can tell the analytic sample for the main analyses using EHRs was10,313. This is stated in the results. This should probably be the figure stated in the abstract. Instead the authors state 1,000,000 in the abstract and 50,000 at the start of the discussion section. The authors should provide a flow chart indicating how the analytic sample is derived and in the abstract and discussion state the analytic sample not the size of the population in the database from which it was derived. Similarly, the authors should consider a flow chart indicating how the final analytic sample for the Corona Virus Survey was derived.

I think the proportion of the sample that has been classified into the three clusters by exploratory network analysis is only 10%. If this interpretation is correct my conclusion would be that this method is not very useful.

The introduction should also include more on the theoretical basis of the difference methods used to Classify PCS in the study and why they might be useful.

Other matters:

Could the authors check that all references are complete. Several including 22, 28 and 38 are not. There may be other references that are incorrect.

The data access statement is vague. The authors say data is available by request form the author, but also that is restricted by the use of special licenses. The authors need to make it much clearer what data is available, and the process required to apply for it. I would be very surprised if the EHR data is available on request, as generally there are many restrictions put in place to limit risk of disclosure. In addition, there is a substantial body of research showing when authors state “data is available on request” in practice it is often very hard if not impossible to get hold of the data.

In the abstract the authors should state the number of ways in which PCS has been defined and given that there does not appear to be very many, list the frequency of PCS symptoms for each method.

Also in the abstract, I found the description of “exploratory network analysis” confusing. Normally, I would associate network analysis with social networks of people, not as a method of investigating clusters and that needs to be clear.

On page 4; Could the authors clarify the source used to derive socio-demographic data. Does it come from the GP records or is it via linkage from a different source?

Page 6; Line 3: Could you please provide a bit more detail on how patients were flagged by GPs as having Covid-19. I am not familiar with the Dutch system so don’t know how this would be done.

Page 6 to 7: There is a single paragraph spanning more than a page. It would be easier to read and understand the different definitions of PCS if this paragraph was broken up into different sections.

Page 7, Line 14: “Individuals were classified as having PCS when they reported at least one symptom, from a selected list of symptoms” If this is the list of symptoms in supplementary table 2 this should be explicitly stated, if it is a different list of symptoms, it should be included as an additional supplementary table. In addition, the n for the number of individuals classified as having PCS is not clear in the text.

The discussion section would be improved if the reasons for a very large difference between PCS and Non-PCS with respect to the proportions working less than 3 months 23% versus 0%) were discussed.

Page 13: I am not sure what the authors mean by

“To our knowledge there are no other studies using a data-driven approach to identify clinical subtypes of PCS.”

The authors cite studies that identify clusters using data, and there are possible other studies e.g. (Bowyer, Huggins et al. 2023) that might be viewed as trying to find subtypes of PCS based on symptoms clustering. Please clarify.

Page 14: There should be more discussion of how representative the GP records really are of the population, perhaps supported by data. The data they have only covers 10% of the population and this may not be fully representative of all groups. Particularly those who are hard to reach. Biases arising from the use of administrative data are not fully appreciated see (Shaw, Harron et al. 2023)

Finally, could you please include in or alongside supplementary table 1 (and possible other tables ) how the categories for age, education, income, and migration status were defined. I note that this is in the text of the main paper, but it would be helpful to have it in the same files as the table as well. Supplementary table 1 was introduced in the text before the variables were described.

Bowyer, R. C. E., C. Huggins, R. Toms, R. J. Shaw, B. Hou, E. J. Thompson, A. S. F. Kwong, D. M. Williams, M. Kibble, G. B. Ploubidis, N. J. Timpson, J. A. C. Sterne, N. Chaturvedi, C. J. Steves, K. Tilling, R. J. Silverwood and C. S. the (2023). "Characterising patterns of COVID-19 and long COVID symptoms: evidence from nine UK longitudinal studies." European Journal of Epidemiology 38(2): 199-210.

Shaw, R. J., K. L. Harron, J. M. Pescarini, E. P. Pinto Junior, M. Allik, A. N. Siroky, D. Campbell, R. Dundas, M. Y. Ichihara, A. H. Leyland, M. L. Barreto and S. V. Katikireddi (2023) "Biases arising from linked administrative data for epidemiological research: a conceptual framework from registration to analyses." European Journal of Epidemiology 37, 1215-1224 DOI: 10.1007/s10654-022-00934-w.

Reviewer #2: 1. In this study, the control group and PCS group were matched based on age, sex, and the COVID-19 period, but the exact method and criteria used for this matching are unclear. In addition, given that similar individuals were matched, how was the extent of matching assessed and validated. It might be helpful to refer to the following paper; PMID: 38802352 (Acute and post-acute respiratory complications of SARS-CoV-2 infection: population-based cohort study in South Korea and Japan).

2. To enhance the clarity of Table 1, it may be beneficial to reorder the results listed. If the primary finding of this table is that there was a significant difference in frequency between the control and the narrow definition group, then it would be better to pair PCS and non-PCS side-by-side rather than in the current sequence.

3. Regarding Figure 1, are the top 10 most prevalent core symptoms defined under the "narrow" and "broad" criteria the same as those mentioned in the figure? Regarding Figure 1, are the top 10 most prevalent core symptoms defined under the "narrow" and "broad" criteria the same as those mentioned in the figure? Could you clarify the criteria used for sorting these symptoms? If the sorting is based on one of the definitions (e.g. narrow definition), it would be helpful to mention this in the method section or consider separating the graphs for the narrow and broad criteria.

4. Figure 2 also requires the same clarification. Please refer to the comment 3.

5. The GP-EHR data and the Corona survey cohort each employed different definitions for classification, yet the findings across these datasets do not align completely. For instance, concluding that "PCS mostly affects middle-aged females" based on the Corona survey cohort does not appear statistically significant. For that finding, was it only for GP-EHR data?

6. The statement in the Results section that "Most symptoms were reported at the same frequency or increased over time (3 to 6 months)" seems somewhat overstated. This observation appears to primarily apply to females.

7. In my understanding, the study concluded on frequency alone, without any adjustment except for matching. It seems to lack a methodology to mitigate potential bias, and even then, an observational study requires a conservative approach to interpreting the findings, which I don't think was considered in this manuscript. It would be valuable to address these considerations more thoroughly in the paper.

6. PLOS authors have the option to publish the peer review history of their article (what does this mean?). If published, this will include your full peer review and any attached files.

Reviewer #1: **Yes: **Richard Shaw

Reviewer #2: No

---

## [Author Response · Author response to Decision Letter 0]

21 Oct 2024

We would like to thank the editor and reviewers for their thorough reading and constructive comments. This has helped us to further improve our manuscript. The reviewers have primarily asked for further clarifications and textual adjustments. In the Cover Letter and the Response to Reviewers document we addressed the comments and questions raised by the editor and reviewers. We hope that the manuscript is now suitable for publication in PLOS ONE.

---

## [Decision Letter · Decision Letter 1]

30 Oct 2024

PONE-D-24-29474R1Comparison of observational methods to identify and characterize Post-COVID syndrome in the Netherlands using electronic health records and questionnairesPLOS ONE

Dear Dr. Bos,

Thank you for submitting your manuscript to PLOS ONE. After careful consideration, we feel that it has merit but does not fully meet PLOS ONE’s publication criteria as it currently stands. Therefore, we invite you to submit a revised version of the manuscript that addresses the points raised during the review process.

We look forward to receiving your revised manuscript.

Kind regards,

Dong Keon Yon, MD, FACAAI, FAAAAI

Academic Editor

PLOS ONE

Additional Editor Comments:

The reviewers have raised a number of very important issues, and their excellent comments will need to be adequately addressed in a revision before the acceptability of your manuscript for publication in the Journal can be determined. We cannot guarantee that your revised paper will be chosen for publication; this would be solely based on how satisfactorily you have addressed the reviewer comments.

Reviewers' comments:

Reviewer's Responses to Questions

**Comments to the Author**

1. If the authors have adequately addressed your comments raised in a previous round of review and you feel that this manuscript is now acceptable for publication, you may indicate that here to bypass the “Comments to the Author” section, enter your conflict of interest statement in the “Confidential to Editor” section, and submit your "Accept" recommendation.

Reviewer #1: (No Response)

Reviewer #2: (No Response)

2. Is the manuscript technically sound, and do the data support the conclusions?

Reviewer #1: Yes

Reviewer #2: Partly

3. Has the statistical analysis been performed appropriately and rigorously? 

Reviewer #1: Yes

Reviewer #2: No

4. Have the authors made all data underlying the findings in their manuscript fully available?

Reviewer #1: No

Reviewer #2: No

5. Is the manuscript presented in an intelligible fashion and written in standard English?

Reviewer #1: Yes

Reviewer #2: Yes

6. Review Comments to the Author

Reviewer #1: The authors have improved the paper considerably. However, there are still a few mistakes and clarifications that need addressing.

Thank you for including the two flowcharts in supplementary figure 1. The flowchart for the Corona Survey Cohort would be improved if it started with the 1,851 individuals who were invited to participate. That way the flowchart would cover a major potential source of bias, non-response.

The inclusion of a discussion of why people with PCS stopped working on page 14 has also improved the paper. However, there is little discussion about the 0% of the non-PCS group stopping working. It is very unusual for nobody in a group to stop working, and I am wondering if this might be due to some form of bias or reverse causation.

Thank you for clarifying that when you stated, “To our knowledge there are no other studies using a data-driven approach to identify clinical subtypes of PCS” in your original submitted manuscript that you are referring to studies based on Electronic Health Records. It was unclear form that draft. However, having clarified to what you are referring to the Bowyer et al 2023 reference may no longer be relevant.

Could you please check ALL your references again. Some of them still do not seem to be complete. In particular, it is not clear what journal the Basharat et al (2022) and Case et al (2022) references are in.There may also be other mistakes.

Finally, while description of the data in the methods section is an improvement. That the data are “somewhat underrepresented like strongly urbanized regions” needs stating in the limitations section, with slightly more fluent English. More than 90 percent of the population of the Netherlands live in cites. The database may be representative of age and sex, but I would be very surprised if it was representative for ethnicity and some measures of socioeconomic disadvantage.

Reviewer #2: 1. In your response, it is mentioned that the control and patient groups were compared to verify the accuracy of the matching. Could you clarify where this is mentioned in the paper and where is the method used to test for no significant differences between the two groups?

2. In order to make a conclusion like this paper “Frequency rates differ between methods and data sources (15-33%) but characteristics of the affected individuals seem less associated as we found that PCS mostly affects middle-aged females”, two parts need to be clearly checked. First, a statistical test between PCS group and non-PCS group. Second, a test for the difference between the broad definition and the narrow definition. Only when the two parts are tested can we conclude that the frequency of PCS differed by definition (first step) and showed less associated characteristics for age and sex (second step). It is true that the mean values for age and frequency for sex seem to be similar, but this can only be concluded with a statistical test, and the results of the paper only showed a test between PSC and non-PCS.

3. In observational studies, matching is a natural approach to address potential biases when examining associations. In this manuscript, however, the statement “we found that PCS mostly affects middle-aged females” seems to rely only on the EHR dataset, despite the Corona Survey Cohort results indicating no significant associations (age [p-value, 0.479], sex [p-value, 0.380]). Given these limitations, the author should at least present statistical test results to support the conclusion, or address these limitation in the manuscript.

4. Please improve the resolution of the figures.

7. PLOS authors have the option to publish the peer review history of their article (what does this mean?). If published, this will include your full peer review and any attached files.

Reviewer #1: No

Reviewer #2: No

---

## [Author Response · Author response to Decision Letter 1]

12 Dec 2024

Manuscript-ID: PONE-D-24-29474

Title: Comparison of observational methods to identify and characterize Post-COVID syndrome in the Netherlands using electronic health records and questionnaires

We would like to thank the reviewers for their thorough reading and constructive comments. This has helped us to further improve our manuscript. Below we addressed the comments and questions raised by the reviewers. We hope that the manuscript is now suitable for publication in PLOS ONE. 

Point-by-point response

Reviewer #1

1. Thank you for including the two flowcharts in supplementary figure 1. The flowchart for the Corona Survey Cohort would be improved if it started with the 1,851 individuals who were invited to participate. That way the flowchart would cover a major potential source of bias, non-response.

We have now added this to the flowchart of the Corona Survey Cohort in Supplementary Figure 1. 

2. The inclusion of a discussion of why people with PCS stopped working on page 14 has also improved the paper. However, there is little discussion about the 0% of the non-PCS group stopping working. It is very unusual for nobody in a group to stop working, and I am wondering if this might be due to some form of bias or reverse causation.

We agree that we did not discuss the 0% yet in the non-PCS group. We have now elaborated on this finding in the Discussion (page 14, lines 13-18) by adding the following text: “In addition, further research which follows a patient group as well as a reference group, in this case the non-PCS group, over a longer period of time and also after the COVID pandemic is important as well. In this current study we only focused on a relatively short period and found very little changes regarding work in the non-PCS group (0% at 3 months and 1% at 6 months) which could be the result of an inclusion bias or related to the fact that ‘experiencing discomfort in daily living’ was part of the definition of PCS.”

3. Thank you for clarifying that when you stated, “To our knowledge there are no other studies using a data-driven approach to identify clinical subtypes of PCS” in your original submitted manuscript that you are referring to studies based on Electronic Health Records. It was unclear form that draft. However, having clarified to what you are referring to the Bowyer et al 2023 reference may no longer be relevant.

Thank you for this clarification. After having read the manuscript by Bowyer et al. 2023 we do feel that it is still relevant to include this in the manuscript. 

4. Could you please check ALL your references again. Some of them still do not seem to be complete. In particular, it is not clear what journal the Basharat et al (2022) and Case et al (2022) references are in. There may also be other mistakes.

Some references where still incomplete because at the time of writing the first draft of the manuscript many publications were still in preprint. We have now carefully checked again all references. 

5. Finally, while description of the data in the methods section is an improvement. That the data are “somewhat underrepresented like strongly urbanized regions” needs stating in the limitations section, with slightly more fluent English. More than 90 percent of the population of the Netherlands live in cites. The database may be representative of age and sex, but I would be very surprised if it was representative for ethnicity and some measures of socioeconomic disadvantage.

We have now rephrased the sentence regarding urbanisation level in the manuscript. More information regarding representativeness of Nivel-PCD can be found in the referred publication: Heins et al. 2023. Nivel-PCD is indeed not representative of ethnicity and socioeconomic status. 

Reviewer #2: 

1. In your response, it is mentioned that the control and patient groups were compared to verify the accuracy of the matching. Could you clarify where this is mentioned in the paper and where is the method used to test for no significant differences between the two groups?

The validation of the matching was not yet mentioned in the manuscript. We have now mentioned this in the manuscript (page 6, lines 20-22). 

2. In order to make a conclusion like this paper “Frequency rates differ between methods and data sources (15-33%) but characteristics of the affected individuals seem less associated as we found that PCS mostly affects middle-aged females”, two parts need to be clearly checked. First, a statistical test between PCS group and non-PCS group. Second, a test for the difference between the broad definition and the narrow definition. Only when the two parts are tested can we conclude that the frequency of PCS differed by definition (first step) and showed less associated characteristics for age and sex (second step). It is true that the mean values for age and frequency for sex seem to be similar, but this can only be concluded with a statistical test, and the results of the paper only showed a test between PSC and non-PCS. In observational studies, matching is a natural approach to address potential biases when examining associations. In this manuscript, however, the statement “we found that PCS mostly affects middle-aged females” seems to rely only on the EHR dataset, despite the Corona Survey Cohort results indicating no significant associations (age [p-value, 0.479], sex [p-value, 0.380]). Given these limitations, the author should at least present statistical test results to support the conclusion, or address these limitation in the manuscript.

As presented in Table 1 we have conducted statistical testing between the PCS group and the non-PCS group for both the narrow and broad definition. In both comparisons we found that the PCS group is older compared to the non-PCS group and more often female. We however do agree that the differences between the PCS and non-PCS group were not significant in the Survey cohort. We therefore rephrased our conclusion in the manuscript (page 16, lines 3-5). 

3. Please improve the resolution of the figures.

We have now improved the resolution of the figures.

---

## [Decision Letter · Decision Letter 2]

7 Jan 2025

PONE-D-24-29474R2Comparison of observational methods to identify and characterize Post-COVID syndrome in the Netherlands using electronic health records and questionnairesPLOS ONE

Dear Dr. Bos,

Thank you for submitting your manuscript to PLOS ONE. After careful consideration, we feel that it has merit but does not fully meet PLOS ONE’s publication criteria as it currently stands. Therefore, we invite you to submit a revised version of the manuscript that addresses the points raised during the review process.

We look forward to receiving your revised manuscript.

Kind regards,

Dong Keon Yon, MD, FACAAI, FAAAAI

Academic Editor

PLOS ONE

Additional Editor Comments (if provided):

Please see my minor comments.

#1. Some of those infected with coronavirus suffer from post-COVID syndrome (PCS) -> Some of those infected with SARS-CoV-2 suffer

#2. n=10.313 -> n=10,313 (comma)

#3. coronavirus in 2021 => SARS-CoV-2 in 2021

#4. Please add the definition of PCS in abstract method section.

#5. without a COVID-19 infection => without a SARS-CoV-2 infection

- Please use the official term carefully.

COVID infection (x) -> SARS-CoV-2 infection

Coronavirus (x) -> SARS-CoV-2

COVID pandemic (x) -> COVID-19 pandemic

#6. In addition literature studies on symptoms that often co-occur do however also often show a cardiorespiratory cluster [42], a generalized-mood cluster [43], and a gastrointestinal cluster [18]

The current authors did not engage in any discussion regarding landmark studies on PCS. In fact, among the cited articles, there are no papers from top-tier journals.

- https://www.nature.com/articles/s41591-022-01689-3

- https://www.nature.com/articles/s41562-024-01895-8

- https://www.nature.com/articles/s41467-023-36223-7

- https://pubmed.ncbi.nlm.nih.gov/38802352/

Thank you

Reviewers' comments:

Reviewer's Responses to Questions

**Comments to the Author**

1. If the authors have adequately addressed your comments raised in a previous round of review and you feel that this manuscript is now acceptable for publication, you may indicate that here to bypass the “Comments to the Author” section, enter your conflict of interest statement in the “Confidential to Editor” section, and submit your "Accept" recommendation.

Reviewer #2: All comments have been addressed

2. Is the manuscript technically sound, and do the data support the conclusions?

Reviewer #2: Yes

3. Has the statistical analysis been performed appropriately and rigorously? 

Reviewer #2: Yes

4. Have the authors made all data underlying the findings in their manuscript fully available?

Reviewer #2: Yes

5. Is the manuscript presented in an intelligible fashion and written in standard English?

Reviewer #2: Yes

6. Review Comments to the Author

Reviewer #2: Thank you for addressing each comment thoroughly and incorporating the revisions into the manuscript. I have no further comments at this time.

7. PLOS authors have the option to publish the peer review history of their article (what does this mean?). If published, this will include your full peer review and any attached files.

Reviewer #2: No

---

## [Author Response · Author response to Decision Letter 2]

10 Jan 2025

In the attached ‘Response to Reviewers’ document we will address the comments and questions raised by the editorial team. We hope that our improved manuscript is now suitable for publication in PLOS ONE. Should you require any further information of have any questions feel free to let us know.

---

## [Editor Report · Decision Letter 3]

14 Jan 2025

Comparison of observational methods to identify and characterize Post-COVID syndrome in the Netherlands using electronic health records and questionnaires

PONE-D-24-29474R3

Dear Dr. Bos,

We’re pleased to inform you that your manuscript has been judged scientifically suitable for publication and will be formally accepted for publication once it meets all outstanding technical requirements.

Kind regards,

Dong Keon Yon, MD, FACAAI, FAAAAI

Academic Editor

PLOS ONE

Additional Editor Comments (optional):

This is an excellent paper!
---

## [Editor Report · Acceptance letter]

17 Jan 2025

PONE-D-24-29474R3 

PLOS ONE

Dear Dr. Bos, 

I'm pleased to inform you that your manuscript has been deemed suitable for publication in PLOS ONE. Congratulations! Your manuscript is now being handed over to our production team.

Kind regards, 

on behalf of

Dr. Dong Keon Yon 

Academic Editor

PLOS ONE